# Regulatory Connections between Iron and Glucose Metabolism

**DOI:** 10.3390/ijms21207773

**Published:** 2020-10-21

**Authors:** Carine Fillebeen, Nhat Hung Lam, Samantha Chow, Amy Botta, Gary Sweeney, Kostas Pantopoulos

**Affiliations:** 1Lady Davis Institute for Medical Research, Jewish General Hospital and Department of Medicine, McGill University, Montreal, QC H3Y 1P3, Canada; carine.fillebeen@mail.mcgill.ca; 2Department of Biology, York University, Toronto, ON M3J 1P3, Canada; nhatl96@my.yorku.ca (N.H.L.); samantha.chow.v@gmail.com (S.C.); amymbotta@gmail.com (A.B.); gsweeney@yorku.ca (G.S.)

**Keywords:** hepcidin, ferroportin, insulin, adipokines, IRP1, IRP2

## Abstract

Iron is essential for energy metabolism, and states of iron deficiency or excess are detrimental for organisms and cells. Therefore, iron and carbohydrate metabolism are tightly regulated. Serum iron and glucose levels are subjected to hormonal regulation by hepcidin and insulin, respectively. Hepcidin is a liver-derived peptide hormone that inactivates the iron exporter ferroportin in target cells, thereby limiting iron efflux to the bloodstream. Insulin is a protein hormone secreted from pancreatic β-cells that stimulates glucose uptake and metabolism via insulin receptor signaling. There is increasing evidence that systemic, but also cellular iron and glucose metabolic pathways are interconnected. This review article presents relevant data derived primarily from mouse models and biochemical studies. In addition, it discusses iron and glucose metabolism in the context of human disease.

## 1. Iron and Energy Metabolism

Iron is a transition metal with critical biological functions [1]. In mammals, most of body iron is present in hemoglobin of red blood cells and mediates oxygen transport. Significant amounts of iron are also present within myoglobin of skeletal muscle cells. Other cell types require smaller quantities of iron for utilization by several metalloproteins. These include metabolic enzymes and oxidoreductases, which catalyze electron transfer reactions. The activity of mitochondrial aconitase, an enzyme catalyzing conversion of citrate to isocitrate in the tricarboxylic acid (TCA) cycle, depends on a 4Fe-4S cluster in its active site. Moreover, four out of five complexes in the mitochondrial electron transport chain contain hemoproteins (such as cytochromes) or iron–sulfur cluster proteins. Thus, iron is essential for cellular energy metabolism.

Cell culture experiments showed that iron depletion inhibits not only mitochondrial aconitase, but also other enzymes of the TCA cycle such as citrate synthase, isocitrate dehydrogenase and succinate dehydrogenase [2]. This decreases formation of NADH and ATP, and also reduces oxygen consumption in the electron transport chain. To compensate for the inhibition in respiration, the iron-depleted cell increases glycolysis for ATP synthesis. On the other hand, excessive iron negatively affects mitochondrial function. Thus, dietary iron overload of mice decreases oxidative phosphorylation in liver mitochondria and also promotes mitochondrial disfunction due to oxidative stress [3]. This is consistent with the notion that while iron is an essential nutrient, it may also become a potent biohazard by promoting oxidative stress [4]. The Janus face of iron indicates that balanced iron metabolism is imperative for health [5]. Mechanisms underlying regulation of systemic and cellular iron metabolism are summarized below. 

## 2. Systemic Iron Metabolism

Developing erythroid cells in the bone marrow and most of cells in other tissues acquire iron from transferrin, the plasma iron carrier [6]. Transferrin is predominantly replenished by iron recycled from tissue macrophages, which phagocytize iron-rich senescent red blood cells and degrade heme via HO-1 (heme oxygenase 1). Liberated iron is then released to the bloodstream via the iron exporter ferroportin for reutilization. Intestinal enterocytes absorb iron from dietary sources via DMT1 (divalent metal transporter 1), which is expressed on the apical site, and release it to plasma on the basolateral site via ferroportin. Luminal iron is previously reduced from Fe^3+^ to Fe^2+^ by ferrireductases, such as DCYTB (duodenal cytochrome B). Ferroportin, DMT1 and DCYTB are transcriptionally induced in iron-deficient enterocytes by HIF2α (hypoxia inducible factor 2α) to stimulate iron absorption [7]. Under physiological conditions, the contribution of dietary iron to the transferrin-bound plasma iron pool is small and mostly serves to compensate for non-specific iron losses.

Iron efflux from cells is critical for body iron homeostasis and is negatively controlled by hepcidin, a peptide hormone that inactivates ferroportin in macrophages, enterocytes and other target cells [8] (Figure 1). Circulating hepcidin is synthesized by hepatocytes in the liver; however, hepcidin is also locally produced in other tissues and appears to have critical cell-autonomous functions [9,10,11]. The hepcidin-encoding *HAMP* gene is primarily induced in response to iron or inflammatory signals [8]. Hepcidin deficiency causes uncontrolled release of iron to plasma, gradual saturation of transferrin and buildup of redox-active non-transferrin-bound iron (NTBI). This is taken up by hepatocytes and other tissue parenchymal cells leading to systemic iron overload (hemochromatosis) [12]. Conversely, sustained inflammatory induction of hepcidin contributes to the anemia of inflammation, the most frequent anemia among chronically ill patients [13].

Iron intake triggers hepcidin induction in response to increased iron saturation of plasma transferrin and secretion of BMP6 (bone morphogenetic protein 6) from liver sinusoidal endothelial cells [14] (Figure 1). Endothelial cells also secrete BMP2, which is thought to control basal hepcidin expression. Binding of BMP6 or BMP2 to cell surface BMP receptors promotes phosphorylation of regulatory SMAD1/5/8, recruitment of SMAD4, and translocation of the complex to the nucleus for transcriptional activation of the *HAMP* promoter. HJV (hemojuvelin) is a BMP co-receptor and enhancer of iron signaling to hepcidin, and its disruption leads to severe hemochromatosis in humans and mice. HFE and TfR2 (transferrin receptor 2) are presumably auxiliary factors, since their inactivation is associated with milder hemochromatosis. HFE physically interacts with TfR1 (transferrin receptor 1) [15], and this may affect its iron signaling function. Iron signaling to hepcidin is negatively regulated by matriptase-2, a serine protease encoded by the *TMPRSS6* gene that operates by cleaving and inactivating HJV.

Inflammatory induction of hepcidin is primarily mediated by IL-6 [16]. Upon binding to its receptor, IL-6 triggers STAT3 phosphorylation by JAK1/2 kinases. Phosphorylated STAT3 translocates to the nucleus for transcriptional *HAMP* induction. Experimental evidence suggests cooperation between BMP/SMAD and IL-6/STAT3 signaling. Thus, pharmacological inhibition of the SMAD pathway [17,18], or genetic inactivation of some of its components [19,20] attenuated inflammatory hepcidin responses.

## 3. Cellular Iron Metabolism

Cellular iron uptake involves binding of transferrin to TfR1 on the plasma membrane, which is followed by internalization of the complex via endocytosis, release of iron from the acidified endosome and exit to the cytosol via DMT1. Internalized iron is utilized in mitochondria for the synthesis of heme and iron–sulfur clusters. Excess iron is either stored in the cytosol within ferritin, an iron storage protein, or exported from cells via ferroportin. The expression of TfR1, ferritin and ferroportin is coordinately regulated by a post-transcriptional mechanism. In iron-deficient cells, iron regulatory proteins (IRP1 and IRP2) interact with iron-responsive elements (IREs) within the untranslated regions of TfR1, ferritin and ferroportin mRNAs [21]. The RNA/protein complexes promote stabilization of TfR1 and translational repression of ferritin and ferroportin mRNAs, leading to enhanced iron acquisition for metabolic purposes. Increased intracellular iron triggers inactivation of IRP1 and IRP2 for IRE-binding, allowing TfR1 mRNA degradation and synthesis of ferritin and ferroportin. These responses prevent excessive accumulation of unshielded iron in the cells. The IRE/IRP system also accounts for the regulation of further proteins directly or indirectly linked to iron metabolism such as DMT1, the heme biosynthetic enzyme ALAS2 (aminolevulinic acid synthase 2), mitochondrial aconitase, or the transcription factor HIF2α.

Even though IRP1 and IRP2 share extensive similarity, they are regulated differently. Thus, iron converts IRP1 to a cytosolic aconitase at the expense of its RNA-binding activity via the assembly of a 4Fe-4S cluster [22]. Contrary to this reversible mode of post-translational regulation, iron triggers proteasomal degradation of IRP2 via the ubiquitin ligase FBXL5 (F-box/LRR-repeat protein 5) [23,24].

## 4. Overview of Glucose Metabolism

Glucose is the principal source for metabolic energy. It is mainly acquired from the diet as breakdown product of food macromolecules but can also be mobilized from glycogen stores or synthesized from other metabolites. Ingested glucose is absorbed by intestinal enterocytes via SGLT1 (sodium–glucose co-transporter 1) [25] and is released to plasma via GLUT2, a member of the GLUT family of facilitative glucose transporters [26]. GLUT transporters also account for cellular uptake of circulating glucose by a passive diffusion mechanism, which is driven by the lower intracellular glucose concentration. Skeletal muscle cells, cardiomyocytes and adipocytes acquire glucose via the insulin-regulated glucose transporter GLUT4, while hepatocytes primarily utilize GLUT2 [27]. Pancreatic β cells take up glucose mostly via GLUT2 in rodents and GLUT1 and GLUT3 in humans [28].

The fate of intracellular glucose differs among cell types. All cells can metabolize glucose via glycolysis to pyruvate, which is further converted to acetyl-CoA to enter the TCA cycle. This key metabolic pathway is prominent in energy-demanding muscle cells. Adipocytes utilize acetyl-CoA for fatty acid biosynthesis to store energy. Hepatocytes mainly convert glucose to glycogen for energy storage and can also synthesize glucose by gluconeogenesis. Glucose uptake by pancreatic β cells is critical for insulin synthesis and systemic glucose regulation.

Plasma glucose levels are maintained within a narrow range by the pancreatic hormones glucagon and insulin. Hypoglycemia triggers secretion of glucagon by pancreatic α cells, which promotes glycogenolysis and gluconeogenesis in the liver, and lipolysis in adipose tissue; these responses aim to restore euglycemia. On the other hand, hyperglycemia triggers secretion of insulin from pancreatic β cells, which promotes glucose uptake for energy production and anabolic processes such as glycogen synthesis and lipogenesis in the liver, muscles and adipose tissue (Figure 2).

Insulin binds to insulin receptors and activates PI3K/Akt signaling cascades [29]. In skeletal muscle cells and adipocytes insulin, signaling promotes translocation of GLUT4-containing storage vesicles to the plasma membrane for glucose absorption. Moreover, in skeletal muscle cells, insulin signaling inactivates the inhibitory GSK3 (glycogen synthase kinase 3), which restores glycogen synthase activity and allows glycogen synthesis. In adipocytes, insulin signaling inhibits HSL (hormone-sensitive lipase) to suppress lipolysis. In hepatocytes, insulin signaling targets GSK3 to induce glycogen synthesis and phosphorylase kinase to inhibit glycogenolysis. In addition, it activates protein synthesis, inhibits gluconeogenesis and stimulates lipogenesis.

## 5. Insights on Iron and Glucose Control from Metabolomics Data

Metabolomics is used to identify and quantify metabolites, small molecules involved as intermediates and products of metabolism. Iron overload or deficiency can both lead to metabolic changes and cause disease. Mass spectrometry is a common method for detection of metabolic signatures in biological samples. Metabolomics offers a powerful tool to identify pathways linking iron and glucose control, potential metabolic links to associated diseases, and biomarkers for diagnosis and prognosis.

Metabolic signatures have been assessed in plasma and livers from mice subjected to dietary iron overload and compared to those of control animals on standard diet [30]. Iron overload promoted an increase in blood glucose, aspartic acid and β-hydroxybutyrate, and a concomitant decrease in blood lactate and malate. This suggests a reprogramming of glucose metabolism and the TCA cycle. In the liver, iron overload resulted in increased glutathione synthesis, presumably to mitigate oxidative stress, and also stimulated the urea cycle. In addition, iron overload was associated with lower plasma and liver carnitine levels, possibly due to its consumption as an antioxidant molecule or as a response to aberrant glucose metabolism.

Another study compared metabolites in serum samples of patients with metabolic syndrome (see below) and iron overload to control subjects, and to metabolic syndrome patients subjected to phlebotomy [31]. Interestingly, the concentrations of sarcosine, citrulline, methioninsulfoxide, and long-chain phosphatidylcholines were significantly altered between the groups, implying that iron may be involved in a multitude of metabolic pathways, some of which have not been previously reported. The changes in long-chain phosphatidylcholines are of particular interest as they were not previously linked to iron homeostasis and were only loosely associated with metabolic dysfunction [32,33,34,35].

## 6. Iron and Glucose Metabolism in Human Disease

Iron overload is an established risk factor for diabetes [36,37]. This is vividly illustrated in the high (20–50%) frequency of diabetes in patients with iron overload disorders such as hereditary hemochromatosis [38] or β-thalassemia [39], which is related to both insulin resistance and destruction of pancreatic β-cells. Moreover, epidemiological studies provided links between aberrant iron metabolism and the metabolic syndrome (MetS), a pathologic condition defined by the combined manifestation of at least three of the following conditions: abdominal obesity, hyperglycemia due to insulin resistance, hyperlipidemia and hypertension. Thus, many MetS patients develop mild systemic iron overload characterized by excess liver iron, the presence of serum non-transferrin bound iron and hyperferritinemia [40,41,42,43]. The combination of unexplained iron overload with insulin resistance is also referred to as dysmetabolic iron overload syndrome (DIOS) and has a prevalence of 15–30% among MetS patients [44]. On the other hand, obesity is also considered as a risk factor for iron deficiency, and some obese patients develop anemia; most likely, these responses are associated with inflammatory induction of hepcidin [45].

Non-alcoholic fatty liver disease (NAFLD) represents the hepatic component of the MetS and constitutes the most frequent liver disease in Western countries [46,47]. It is characterized by excessive fat deposition in hepatocytes (steatosis) in the absence of other causes of liver injury (i.e., alcohol abuse or viral hepatitis). In many NAFLD patients, liver disease progresses from simple steatosis to non-alcoholic steatohepatitis (NASH), a chronic inflammatory condition that may further lead to liver fibrosis, cirrhosis and hepatocellular carcinoma (HCC). Excess hepatic iron is a risk factor for progression of NAFLD to NASH, liver cirrhosis and HCC [48]. Thus, liver iron deposition is more frequent in individuals with NASH-related cirrhosis with HCC than in HCC-free controls [49]. Consequently, manipulation of iron metabolic pathways has been proposed to offer a promising therapeutic target [50]. 

In some occasions, interventions leading to reduction of iron stores (phlebotomy, treatment with iron-chelating drugs or use of iron-deficient diets) have improved insulin sensitivity in MetS patients [51,52] and rodent models [53,54]. However, clinical data on iron depletion strategies to improve insulin sensitivity or ameliorate metabolic liver disease have had mixed outcomes and are largely inconclusive [36,55]. Therefore, a better understanding of the complex mechanisms linking iron and glucose metabolism at the systemic and cellular levels is required to develop and improve iron-related therapeutic interventions.

## 7. Links Between Systemic Iron and Glucose Metabolism

Levels of both plasma iron and glucose are subjected to negative hormonal regulation by hepcidin and insulin, respectively [56]. This analogy indicates that iron and glucose metabolism may be interconnected. In fact, there are several examples supporting this view. Thus, insulin directly induces hepcidin expression in hepatocytes via STAT3; this response is attenuated and may contribute to iron overload in a rat model of type 2 diabetes induced by combination of streptozotocin treatment and high-fat diet [57]. Likewise, glucose intake was shown to augment circulating hepcidin and decrease serum iron levels in healthy volunteers [58]. It is unclear whether this was a result of systemic hepcidin induction in hepatocytes or local hepcidin production by insulin-secreting pancreatic β-cells. In vitro studies showed that glucose stimulates hepcidin expression in rat INS-1E insulinoma but not human HepG2 hepatoma cells [58]. Another study found that high, glucotoxic glucose levels rather suppress hepcidin in islets from db/db mice, which lack the receptor of the insulin-sensitizing hormone leptin, and in mouse Min6 insulinoma cells [59]. This response appears to promote iron overload, oxidative stress, mitochondrial depolarization and β-cell dysfunction [60]. The underlying mechanisms are not well understood; nevertheless, the metabolic phenotypes of obese leptin receptor-deficient db/db [60], leptin-deficient ob/ob [54] and KKAy [61] mice can be improved by dietary iron restriction.

Metabolic cues can also modulate systemic hepcidin expression in the liver. Thus, mice respond to starvation by transcriptional induction of Hamp mRNA, possibly to preserve sufficient iron levels in tissues with high metabolic needs [62]. The mechanism involves the transcription factor CREBH and the transcriptional coactivator PPARGC1A, which are activated by gluconeogenic signals. In another example, leptin was reported to increase HAMP mRNA expression in human Huh7 hepatoma cells via STAT3 [63]. In line with this finding, ob/ob and db/db mice exhibit lower serum hepcidin levels [64,65], while administration of recombinant leptin to ob/ob mice increased hepatic Hamp mRNA and serum hepcidin [64]. It should also be noted that wild-type mice on obesogenic high-fat diets exhibit reduced Hamp mRNA expression and tend to develop mild iron deficiency [66,67]. Hepcidin-independent suppression of iron absorption is another contributing factor [68]. On the other hand, high-fat diets have been shown to promote Hamp mRNA induction and iron overload in rats [69,70]. The upregulation of Hamp mRNA is consistent with the fact that obesity is associated with chronic low grade inflammation and release of hepcidin-inducing inflammatory cytokines such as IL-6 by adipocytes [71,72]. Nevertheless, the reasons for the discrepancies between mouse and rat models are unclear.

Tmprss6-/- mice express high levels of hepcidin due to disruption of the *Tmprss6* gene encoding the hepcidin inhibitor matriptase-2 [73]. These animals, which offer a model of iron-refractory iron deficiency anemia, are protected against high-fat diet-induced obesity and liver steatosis [74]. Moreover, they exhibit improved glucose tolerance and insulin sensitivity, enhanced fat lipolysis and reduced adipocyte hypertrophy. This phenotype was reversed upon injection of an anti-HJV antibody blocking hepcidin overexpression and was enhanced by injection of iron dextran that further stimulates hepcidin. Interestingly, iron dextran also improved metabolism in wild-type mice via hepcidin induction [74]. Considering that iron overload is generally associated with metabolic dysfunction (see below), this finding appears paradoxical. Nevertheless, feeding wild-type or hepcidin-deficient Hjv-/- mice with a combination of high-iron and high-fat diet was also protective against obesity, liver steatosis, hyperglycemia and hypercholesterolemia triggered by the high-fat diet alone [66]. These data underline the complexity of metabolic responses to alterations in hepcidin and systemic iron levels. It is conceivable that additional factors can shift the balance towards improved or worsened metabolic function when hepcidin and systemic iron levels fluctuate.

One potential factor is the intestinal microbiota. Under physiological symbiotic conditions, the gut is the source of microbial-derived nutrients and metabolites, such as short-chain fatty acids (SCFA) and secondary bile acids, which cross the intestinal barrier and reach the liver to control metabolic pathways. However, intake of high-fat and low-fiber diets promotes dysbiosis, inflammation and changes in microbiome composition, favoring propagation of pathogenic bacteria. In this setting, alterations in the intestinal barrier allow increased absorption of branched chain amino acids (BCAA) and lipopolysaccharide (LPS) at the expense of SCFA and bile acids. These responses contribute to peripheral insulin resistance, immunological dysfunction and liver steatosis [75,76,77]. Iron is emerging as an important regulator of the microbiome [78]. While intestinal bacteria are known to utilize luminal dietary iron for growth, it appears that they also respond to tissue iron levels. Thus, variations in microbiome composition have been reported in humans [79] and mice [80] receiving parenteral iron, and in mice with genetically altered iron homeostasis [81]. Conversely, intestinal bacteria can control dietary iron absorption and iron homeostasis in the host by secreting metabolites modulating expression of HIF2α in enterocytes [82].

Another factor that may affect the interplay between systemic iron and glucose metabolism is the iron content of adipose tissue, which can modulate synthesis of insulin-regulating adipokines (Figure 3). Thus, adipocyte iron induces expression of the insulin-inhibitory adipokines resistin [83] and RBP-4 (retinol-binding protein 4) [84], causing insulin resistance. Experiments in 3T3-L1 adipocytes and mice showed that iron also inhibits expression of insulin-sensitizing leptin via phosphorylation-dependent inactivation of CREB [85]. In addition, systemic iron overload suppresses adiponectin, another insulin-sensitizing adipokine, by a mechanism involving the FOXO-1 (forkhead box protein O1) transcription factor [86]. However, Hfe-/- mice, a model of hereditary hemochromatosis, exhibited increased adiponectin expression and improved glucose tolerance. This was attributed to reduced iron content of adipose tissue, in spite of systemic iron overload [86]. Mechanistically, hepcidin deficiency would allow unrestricted ferroportin expression and iron efflux from adipocytes. Nevertheless, further experiments with adipocyte-specific ferroportin knockout mice were not supportive to this view [87], emphasizing the need for more studies to clarify the disparity. This could be related to genetic background differences in mouse strains, which is known to influence iron content and function of adipose tissue [88].

Adipose tissue iron may be affected by additional signals. One of them is locally produced adipose tissue hepcidin, which has been documented in obese humans [89] and mice [90]. Additionally, there is evidence that a subset of M2-like anti-inflammatory resident macrophages with high iron storage capacity is present in the healthy adipose tissue and serves to spare adipocytes from iron overload. However, in obesity, this subpopulation acquires M1-like pro-inflammatory features and loses its iron storage capacity, favoring adipocyte iron overload [91]. This important topic is extensively discussed in a comprehensive review article [92].

## 8. Links between Cellular Iron and Glucose Metabolism

At the cellular level, insulin was reported to stimulate TfR1-mediated iron uptake in primary rat adipocytes [93] and in human HepG2 hepatoma cells [94]. Importantly, excessive cellular iron accumulation has been associated with insulin resistance in primary rat adipocytes [95], primary mouse hepatocytes [96] or rat H9c2 cardiomyocytes [97]. Conversely, pharmacological iron chelation induced glucose uptake and enhanced insulin sensitivity in HepG2 cells and in the rat liver [53]. These responses were linked to transcriptional induction of the glucose transporter GLUT1 and the insulin receptor InsR following stabilization of HIF1α (hypoxia-inducible factor 1α), which is sensitive to iron-dependent degradation [53]. Iron chelation was also shown to induce GLUT1 expression in rat L6 muscle cells [98]. On the other hand, iron depletion, but also iron overload negatively affected differentiation of human 3T3-L1 adipocytes, underlying the importance of balanced iron metabolism.

Intracellular iron distribution appears to be critical for metabolic functions, at least in some cell types. This became evident with the identification of the outer mitochondrial membrane iron–sulfur cluster protein mitoNEET, a target of the antidiabetic drug pioglitazone [99], as an inhibitor of mitochondrial iron import [100]. Thus, overexpression of mitoNEET preserves insulin sensitivity and reduces oxidative stress in adipocytes, even though it causes expansion of adipose tissue; moreover, it stimulates expression of adiponectin [100]. It appears that mitoNEET, which is stabilized by the antidiabetic drug pioglitazone [99], has pleiotropic functions and is also involved in the control of energy metabolism, iron–sulfur cluster biogenesis and cell proliferation [101]. Interestingly, tetracycline-inducible overexpression of mitoNEET in pancreatic β cells caused glucose intolerance in a mouse model due to activation of a mitophagic pathway [102]. On the other hand, mitoNEET induction in pancreatic α cells elicited protective effects on β cells [102]. 

Iron is important for proper function of pancreatic β cells (Figure 4); however, excessive iron accumulation has been shown to impair insulin secretion in various settings [60,103,104]. Experiments with rat islets and INS-1E cells identified DMT1 as a key mediator of iron overload and β cell dysfunction following its induction by the inflammatory cytokine IL-1β. Moreover, mice bearing β cell-specific ablation of DMT1 were protected against glucose intolerance in models of type 1 and type 2 diabetes induced by either low-dose streptozotocin treatment or a high-fat diet, respectively [105]. Interestingly, these mice also exhibit defects in glucose-stimulated insulin secretion without inflammatory stimuli, highlighting the importance of physiological iron content in β cell function, but also the critical role of DMT1 in β cell iron supply.

Surprisingly, mice with genetic hepcidin deficiency (Hamp-/-) [106] or resistance (Slc40a1^C326S/C326S^) [107] do not exhibit defects in insulin production, in spite of pancreatic iron overload. The reason is that in these mouse models of hemochromatosis, and contrary to human hemochromatosis patients, excess iron accumulates in pancreatic acinar but not β cells. This finding is consistent with the improved insulin sensitivity that has been documented in Hfe-/- mice [86].

Moreover, Hfe-/- mice exhibit improved basal (not insulin-stimulated) glucose uptake in skeletal muscles, in spite of a two-fold increase in iron content of this tissue compared to wild-type controls. This has been attributed to increased phosphorylation of AMPK (AMP-dependent kinase) [108]. However, excessive iron overload appears detrimental for the function of skeletal muscle cells and leads to insulin resistance [109]. Several mechanisms may account for this. Thus, endoplasmic reticulum (ER) stress has been proposed to promote upregulation of TfR1 leading to increased iron uptake and insulin resistance in human skeletal muscle cells, which could be prevented by iron chelation or TfR1 knockdown [110]. Similarly, TfR1-mediated iron overload was shown to be critical for palmitate-induced insulin resistance in the same cell model [111]. Autophagy plays a crucial role in skeletal muscles’ response to insulin via the Akt pathway [112]. In this context, iron overload was observed to perturb insulin signaling by impairing autophagy processes [113].

Interestingly, there appears to be distinct cellular repercussions between acute and prolonged exposures to iron overload. Hence, short-term iron overload stimulated autophagy induction through inhibition of the mammalian target of rapamycin complex 1 protein (mTORC1), conceivably as a compensatory measure against iron overload-related stress. Optimal autophagic flux, however, requires proper fusion of autophagosomes with lysosomes to form autolysosomes. Following degradation of autophagic materials, autolysosomal membranes must be resorbed into the endomembrane system in order to regenerate functional lysosomes, a newly discovered process known as autophagic-lysosome regeneration (ALR) [114]. Chronic iron overload eventually impaired ALR, causing accumulation of autolysosome while depleting free lysosomes and effectively halting autophagic flux [113]. This effect was mediated by the loss of mTORC1 reactivation, followed by insulin resistance and decreased Akt phosphorylation. Both ALR and insulin sensitivity were recovered upon iron withdrawal. Furthermore, forced mTORC1 activation not only prevented autolysosome accumulation but also recovered autophagic flux and insulin sensitivity under iron overload. This evidence suggests that skeletal muscle iron overload specifically induces insulin resistance by inhibiting ALR and autophagic flux.

## 9. IRP1 and Glucose Metabolism

Irp1-/- mice were initially reported to lack any discernible phenotype [115] but were subsequently shown to develop erythrocytosis and pulmonary hypertension as a result of translational de-repression of HIF2α mRNA [116,117,118]. Accumulation of HIF2α is accompanied by transcriptional induction of downstream targets such as erythropoietin and endothelin 1, which account for the pathological phenotypes. Preliminary data demonstrated that Irp1-/- mice also exhibit hypoglycemia and improved glucose clearance in oral glucose tolerance tests [119]. The underlying mechanism is not clear, but it is tempting to speculate a link with HIF2α, considering that this transcription factor enhances insulin sensitivity [120,121] but also represses glucagon signaling for gluconeogenesis [122]. Another possibility is that the lack of IRP1 perturbs adipose tissue function, because the cytosolic aconitase activity of IRP1 appears essential for sustaining adipogenic capacity [123]. These findings establish IRP1 as a potential regulator of glucose metabolism.

Further evidence is provided by data obtained in a *Drosophila melanogaster* model. A genetic screen using drosophila larvae revealed that IRP-1A physically and genetically interacts with AGBE (1,4-alpha-glucan branching enzyme), which catalyzes the addition of branches to growing glycogen during glycogen synthesis [124]. The physical interaction is preserved between human IRP1 and GBE1, the orthologs of drosophila IRP-1A and AGBE, respectively. Importantly, IRP-1A only interacts with AGBE in iron-replete cells and in the presence of the Cisd2 protein, the drosophila homologue of mitoNEET. Under these conditions, IRP-1A (and mammalian IRP1) assembles a 4Fe-4S cluster and operates as cytosolic aconitase at the expense of its IRE-binding activity. Earlier biochemical studies showed that mitoNEET can repair the 4Fe-4S cluster of IRP1 [125].

Huynh et al. demonstrated that AGBE promotes nuclear translocation of IRP-1A, where Cisd2 maintains its 4Fe-4S cluster [124]. This occurs in the prothoracic gland of drosophila larvae, a tissue with high iron requirements. Prothoracic gland iron is used as cofactor of iron-containing enzymes involved in the synthesis of α-ecdysone, a steroid hormone crucial for development. Importantly, nuclear [4Fe-4S]-IRP-1A binds to histones and acts as transcriptional repressor of steroidogenic genes. In addition, AGBE remains inactive, favoring glucose catabolism and energy production via the TCA cycle and oxidative phosphorylation. Conversely, under iron deficiency, AGBE gets dissociated from IRP-1A and becomes active, favoring storage of glucose into glycogen (or catabolism via glycolysis to lactate). While the IRP-1A/AGBE interaction was identified using the prothoracic gland of drosophila larvae, proteomic studies using whole drosophila larvae showed that IRP-1A also interacts with glycogen synthase, a key enzyme of glycogen synthesis [124]. These data provide further evidence for a coordinate regulation of iron and glucose metabolism in the drosophila model. Possible implications for mammalian iron and glucose metabolism are discussed in an excellent review article [126].

## 10. IRP2 and Glucose Metabolism

Irp2-/- mice develop microcytic anemia [127,128] and age-dependent neurological defects with variable penetrance [129,130,131,132]. Recent data suggested that these animals also exhibit glucose intolerance and develop diabetes [133]. This phenotype is caused by functional iron deficiency in pancreatic β cells due to reduced TfR1 expression and excessive ferritin accumulation as a result of IRP2 ablation. Iron deficiency impairs iron–sulfur cluster biogenesis and thereby inhibits the activity of Cdkal1, a 4Fe-4S cluster-containing enzyme that catalyzes the methylthiolation of t6A37 in cytosolic tRNA^Lys^_UUU_. This leads to defective lysine incorporation in proinsulin and reduced production of functional insulin. Iron supplementation normalizes proinsulin secretion and insulin levels in Irp2-/- mice, islets from these animals and rat INS-1 832/13 insulinoma cells subjected to CRISPR/Cas9-mediated knockout of IRP2. Taken together, these findings highlight the importance of proper iron metabolism in pancreatic β cells (Figure 4) and raise the possibility for a link between IRP2 and diabetes in humans.

## 11. Conclusions

Both iron and glucose are essential for cellular energy production: the former as a component of key metabolic enzymes and the latter as the principal energy source. Clinical studies suggested that deregulation of iron metabolism in iron overload disorders is associated with metabolic dysfunction. Moreover, deregulation of glucose homeostasis in the metabolic syndrome often correlates with iron overload. Nevertheless, attempts to target iron pathways in order to improve metabolic functions have had limited success thus far. This may be related to a knowledge gap on mechanisms linking iron and glucose homeostasis. Herein, we presented data, mainly obtained from mouse models, suggesting an interplay between systemic iron and glucose homeostasis involving their hormonal regulators hepcidin and insulin, respectively. The metabolic phenotypes of Irp1-/- and Irp2-/- mice identified iron regulatory proteins, IRP1 and IRP2, as potential metabolic regulators. Experiments in the drosophila model demonstrated an additional unexpected function of the drosophila IRP1 orthologue in regulation of glycogen synthesis; it remains to be explored whether this pathway is conserved in mammals. A deeper understanding of the molecular mechanisms linking iron and glucose metabolism may pave the way for the identification of new pharmacological targets and the development of relevant therapeutic interventions for the treatment of common metabolic disorders.

## Figures and Tables

**Figure 1 ijms-21-07773-f001:**
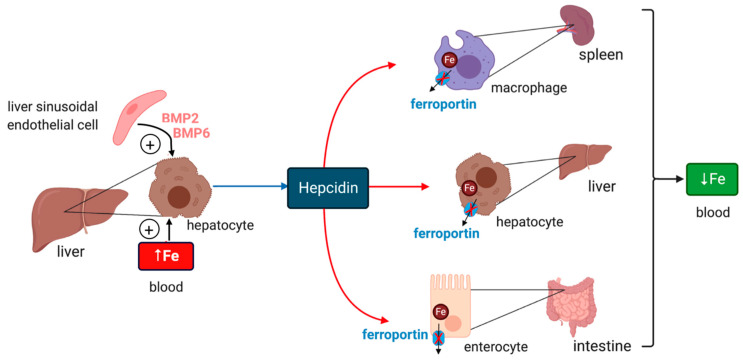
Hormonal regulation of systemic iron traffic by hepcidin. Hepcidin is synthesized in hepatocytes of the liver in response to hyperferremia iron or secretion of bone morphogenetic protein (BMP6) and BMP2 from liver sinusoidal endothelial cells; BMP6 secretion reflects increased body iron stores. It binds to the iron exporter ferroportin in target cells (red arrows) such as tissue macrophages, hepatocytes and intestinal epithelial cells and inhibits ferroportin-mediated iron efflux. Hepcidin binding directly inhibits iron efflux from ferroportin and also promotes ferroportin internalization and degradation. These responses cause cellular iron retention and reduce plasma iron levels.

**Figure 2 ijms-21-07773-f002:**
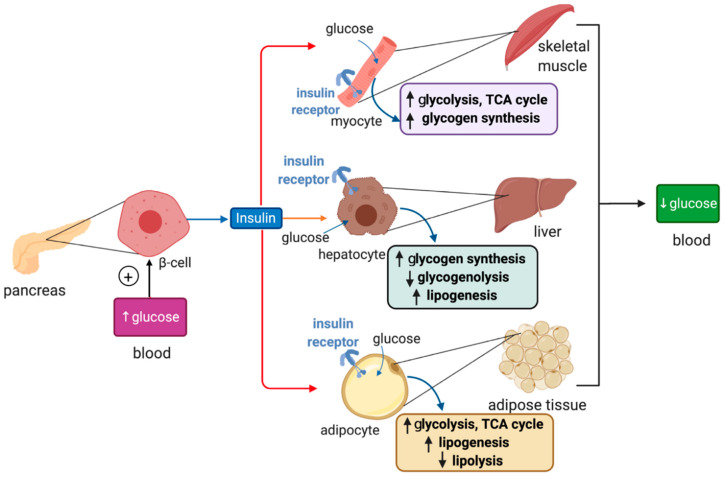
Hormonal regulation of glucose metabolism by insulin. Insulin is synthesized in pancreatic β cells in response to hyperglycemia. It binds to insulin receptors in target cells (red arrows) such as skeletal muscle cells, hepatocytes and adipocytes and induces signaling pathways that promote glucose uptake, catabolism or storage. This reduces plasma glucose levels.

**Figure 3 ijms-21-07773-f003:**
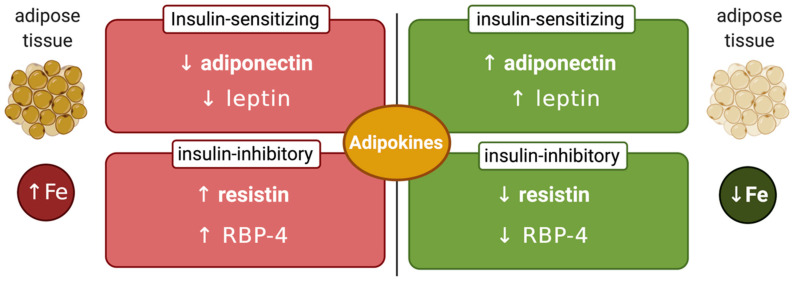
Iron-dependent regulation of adipokines. Increased adipose tissue iron levels inhibit insulin-sensitizing adiponectin and leptin and stimulate insulin-inhibitory resistin and retinol-binding protein 4 (RBP-4). Conversely, reduced adipose tissue iron levels stimulate insulin-sensitizing adiponectin and leptin and inhibit insulin-inhibitory resistin and RBP-4.

**Figure 4 ijms-21-07773-f004:**
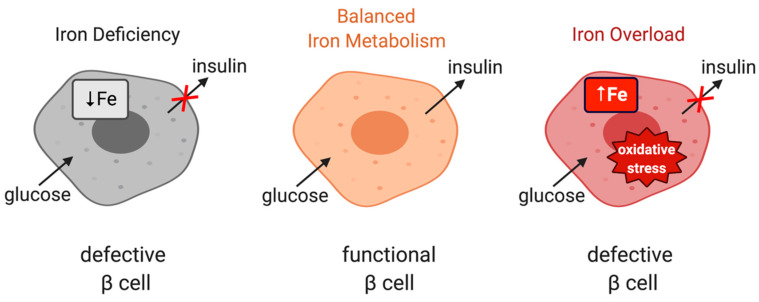
The role of iron in the physiological function of pancreatic β cells. Balanced iron metabolism is critical for proper glucose-stimulated insulin production. Iron deficiency as well as iron overload impair insulin expression.

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
