# Peer review of "Regulatory Connections between Iron and Glucose Metabolism"

_ijms, 2020, doi:10.3390/ijms21207773_

Round 1
Reviewer 1 Report
This review describes parallel changes of glucose and iron metabolism as well as insulin and hepcidin modulation and investigates possible molecular links at the level of organisms and cells.
General: This is a well-written, innovative, and timely review on a topic with which the authors are quite familiar. However, there are a few inaccuracies and unclear issues that need to be corrected. Moreover, the structure of the manuscript would benefit from some rearrangements and additions.
Specific:
- Figure 1: The meaning of the red arrows is confusing. Stimulatory? Inhibitory? I think different types of arrowheads for both mechanisms would be helpful.
- Line 120: The distribution of glucose uptake transporters (I assume SGLT-1 is relevant in this context) in (human) small intestine is not limited to jejunum. Check for instance Lehmann A, Hornby PJ.Intestinal SGLT1 in metabolic health and disease. AmJ Physiol Gastrointest Liver Physiol 310: G887–G898, 2016 for clarification and correct accordingly.
- Line 125: GLUT2 is not the relevant efflux transporter in humans. Check for instance GLUT2 (SLC2A2) is not the principal glucose transporter in human pancreatic beta cells: implications for understanding genetic association signals at this locus by McCulloch LJ, van de Bunt M, Braun M, Frayn KN, Clark A, Gloyn AL. Mol Genet Metab. 2011 Dec;104(4):648-53 for clarification and correct accordingly.
- Lines 144-151. It should be mentioned that insulin not only regulates GLUT4-mediated glucose uptake in skeletal muscle cells but also in adipocytes (textbook knowledge!).
- Lines 155-239. The reviewer wonders whether the authors have made it sufficiently clear that obesity is linked to inflammation, even that adipocytes release substantial amounts of cytokines, including IL-6, which directly affects hepcidin release (see line 87). Check for instance The Pathogenesis of Obesity-Associated Adipose Tissue Inflammation. Engin A. Adv Exp Med Biol. 2017; 960:221-245, or The role of adipocytes in the modulation of iron metabolism in obesity by Coimbra S, Catarino C, Santos-Silva A. Obes Rev. 2013 Oct;14(10):771-9. The implications should be emphasized in a more focused manner.
- Legend to Figure 4: The 2nd sentence should likely read “Balanced iron metabolism is critical…”, rather than “Balanced is critical…”.
- Chapters 9 and 10 describe whole-body correlative phenomena without much mechanistic information. They could be moved after Chapter 4 and be followed by the more mechanistic Chapters 5-8 of the current version that provide information on causal relationships.
- The manuscript would benefit from a summarizing final chapter that draws conclusions on the current knowledge in the field and an outlook hinting at future perspectives.
Author Response
We thank reviewer 1 for his/her insightful comments and great suggestions. A point-by-point response to specific issues is provided below. All modifications in the revised manuscript are highlighted in red.
- We modified Figure. 1 (and Figure. 2) indicating stimulatory effects with a + symbol. We also updated the legend to Figure. 1 for clarity.
- We updated this section and added the new reference as suggested (new lanes 121-122).
- We updated the statement on GLUT transporters in pancreatic beta cells and added the new reference as suggested (new lanes 127-128).
- It was already mentioned in the first version of the manuscript that insulin not only regulates GLUT4-mediated glucose uptake in skeletal muscle cells but also in adipocytes; this statement can now be found in new lane 148.
- Following the reviewer’s suggestion, we added a new sentence on obesity and inflammation in new lanes 242-244. We also added the suggested references.
- We corrected the legend to Figure 4.
- We rearranged previous chapters 9 and 10 as recommended (now they are chapters 5 and 6, respectively).
- We added a final “Conclusions” chapter as suggested by the reviewer.
Reviewer 2 Report
In the manuscript “Regulatory connections between iron and glucose metabolism”, the authors present relevant data from mouse models and biochemical studies, and discuss iron and glucose metabolism in human disease. This manuscript is very interesting and may contribute to the knowledge in the field of life science.
Author Response
We thank reviewer 2 for his/her positive comments.
Round 2
Reviewer 1 Report
- We modified Fig. 1 (and Fig. 2) indicating stimulatory effects with a + symbol. We also updated the legend to Fig. 1 for clarity. Ok.
- We updated this section and added the new reference as suggested (new lanes 121-122). Lines 139-141 are meant. The references (25 and 26) are incorrect or mixed up.
- We updated the statement on GLUT transporters in pancreatic beta cells and added the new reference as suggested (new lanes 127-128). Lines 144 and 161-162 (??) are meant. The references (27 and 28) are mixed up or incorrect.
- It was already mentioned in the first version of the manuscript that insulin not only regulates GLUT4-mediated glucose uptake in skeletal muscle cells but also in adipocytes; this statement can now be found in new lane 148. Ok.
- Following the reviewer’s suggestion, we added a new sentence on obesity and inflammation in new lanes 242-244. We also added the suggested references. Ok.
- We corrected the legend to Figure 4. Ok.
- We rearranged previous chapters 9 and 10 as recommended (now they are chapters 5 and 6, respectively). Ok.
- We added a final “Conclusions” chapter as suggested by the reviewer. Ok.
Author Response
We apologize for the confusion with the lines and references. It was caused by a technical glitch during formatting of the references: Endnote added a second reference list at the end of the document without deleting the previous one. This problem is now solved and the reference numbers have been corrected.